# Epistemic trust and associations with psychopathology: Validation of the German version of the Epistemic Trust, Mistrust and Credulity-Questionnaire (ETMCQ)

**Anna-Maria Weiland**[1], **Svenja Taubner**[2], **Max Zettl**[2], **Leonie C. Bartmann**[3], **Nina Frohn**[3], **Mirijam Luginsland**[4], **Jana Volkert**[5]*

1 Department of Clinical Psychology, Psychological University Berlin, Berlin, Germany, 2 Institute for Psychosocial Prevention, Medical Faculty, Center for Psychosocial Medicine, University Hospital Heidelberg, Heidelberg, Germany, 3 Institute of Psychology, Karl-Ruprechts-University Heidelberg, Heidelberg, Germany, 4 Department Psychology, University of Kaiserslautern-Landau, Landau, Germany, 5 Clinic for Psychosomatic Medicine and Psychotherapy, University of Ulm, Ulm, Germany

* jana.volkert@uni-ulm.de

**Data Availability Statement:** All relevant data files are available from the PsychArchives database

## Abstract

Epistemic trust, defined as trust in socially transmitted knowledge, is discussed as a psychopathological factor in the context of new transdiagnostic approaches for the assessment of mental disorders. The aim of this study is to test the factorial, convergent, and discriminant validity of the German version of the new Epistemic Trust, Mistrust and Credulity–Questionnaire (ETMCQ). Data were collected cross-sectionally from the German-speaking general population ($N = 584$) and in a second sample of clinical ($n = 30$) and non-clinical ($n = 30$) participants. The previously proposed three-factor structure of the ETMCQ was analyzed using confirmatory factor analysis. The ETMCQ's ability to differentiate between clinical and non-clinical participants was tested with $t$-tests. Correlations with early childhood trauma, maladaptive personality traits, and impairments in personality functioning were examined. The relationship between epistemic trust and mentalization was analyzed in a structural equation model. Regarding the factorial validity, the model fit of the originally proposed ETMCQ proved to be insufficient. The model fit to the data was good for a shortened 12-item version. The study was unable to identify any significant differences between clinical and non-clinical participants. For mistrust and credulity, correlations with associated constructs supported their construct validity. However, the results for the trust subscale were heterogeneous. The study offers initial empirical support for a revised 12-item self-report measure of epistemic trust and for the link between mistrust and credulity with markers of psychopathology. Further investigation of the ETMCQ and its psychometric properties, as well as research on integration of epistemic trust into new, transdiagnostic approaches to psychopathology is needed.

(https://doi.org/10.23668/psycharchives.8215) or are within the manuscript and its Supporting Information files.

**Funding:** The data collection of the study was funded by the Leibniz-Institute for Psychology (ZPID; https://leibniz-psychology.org/). The funders had no role in study design, data collection and analysis, decision to publish, or preparation of the manuscript.

**Competing interests:** The authors have declared that no competing interests exist.

## Introduction

Epistemic trust (ET) refers to an individual's willingness to perceive information transmitted by others as relevant, trustworthy, and generalizable to other contexts [1] and provides the basis for adaptive social learning and successful action in the social environment [2, 3].

The concept of ET has been described as a multidisciplinary theoretical model of resilient social functioning and learning [4, 5], where one adaptive and two maladaptive stances can be identified: the adaptive and functional stance of trust, and the two maladaptive stances of mistrust and credulity [6]. The functional stance of trust requires flexible and adequate vigilance. This is essential for adapting to new social situations and to learn about the world and oneself, and safeguarding against misinformation and manipulation [4, 5].

Epistemic mistrust shows in assuming malevolent intentions and motives behind the actions of others. This may result in an impaired ability to modify or update one's views through social learning [3]. These individuals tend to be inflexible in interpersonal contact, frequently report feelings of loneliness and isolation and may be at an elevated risk of developing psychopathology [3]. The second maladaptive stance, epistemic credulity, is characterized by information to be relevant and accurate without sufficient scrutiny [7]. Credulity renders a person vulnerable to misinformation, manipulation and exploitation [5].

The emergence of ET, mistrust, or credulity is closely linked to other important concepts of developmental psychology including attachment, early childhood trauma, mentalizing, and personality functioning. ET and mentalizing, the ability to imagine mental states (e.g., desires, emotions, and motives) in oneself and others [8] develop in secure attachments to sensitive caregivers [1, 2]. The significance of the interactions between the infant and their caregiver for ET is described in the theory of "natural pedagogy" by Csibra and Gergely [9, 10], which posits that infants are innately sensitive to signals transmitted by others and adapt their learning behaviors accordingly [9, 10, 11]. These signals, referred to as ostensive cues (e.g., direct eye contact, name calling, eyebrow raising, nurse talk), open a channel of communication and render infants receptive to social learning. In other words: sensitive caregivers use effective mentalizing, produce ostensive cues and this serves as a catalyst for functional ET [3, 4].

However, these constructs also have a reciprocal influence—for instance, epistemic mistrust and credulity can impair mentalizing ability. Overall, mentalization and epistemic stance are interrelated in complex ways [12]. Initial empirical evidence supports the primarily theoretical assumptions by indicating positive associations between both maladaptive epistemic stances, mistrust and credulity, and difficulties in understanding mental states [6, 13–15].

Regarding the association between early childhood trauma and epistemic stance, the development of both epistemic mistrust and credulity is considered to be an adaptive response to a damaging social environment [4]. Initial empirical research indicates a low to moderate positive association between adverse childhood experiences, such as neglect and abuse, and both epistemic mistrust and credulity, as well as small negative associations with ET [6, 13, 14, 16, 17].

The ET framework has also recently been discussed as a general factor in psychopathology [4]. According to Fonagy et al. [4, 5], the emergence of psychopathology, specifically personality disorders, is understood as an evolutionary form of adaptation to an adverse social environment and a lack of ET as a mediating link in between. However, these assumptions are primarily theoretical. In relation to severe mental disorders, such as borderline personality disorder, there is preliminary evidence for an etiological influence of the epistemic stance [18]. A cross-sectional analysis revealed that individuals with a positive screen for borderline personality disorder exhibited higher levels of mistrust and credulity than those with a negative screen [15]. Similarly, the positive associations between the maladaptive epistemic stances (mistrust

and credulity), and overall symptom severity indicate the same direction [6, 14, 19]. In addition, Kampling et al. [16] address the empirical research gap on the link between epistemic stance and psychopathology with initial findings on the predictive influence of reduced ET on the level of personality functioning. Personality functioning and maladaptive traits are operationalized as transdiagnostic markers of psychopathology in the new dimensional classification systems in the ICD-11 [20] and the Alternative Model of Personality Disorders (AMPD) in Section III of the DSM-5 [21]. To the best of our knowledge, the empirical investigation of associations between epistemic stances and psychopathology markers such as Criterion A (impairment of personality functioning) and Criterion B (maladaptive traits) of the AMPD is still lacking.

## Development of the ETMCQ

In the past, various approaches to measuring ET have been undertaken. These included experimental designs such as Trust Games [22, 23] and the Epistemic Trust Assessment [24], which are impractical for large samples because of their time- and resource-intensiveness. Also, the Inventory of Parent and Peer Attachment [25, 26], a self-report questionnaire more commonly used to measure ET, was found to be too nonspecific [18, 27, 28]. Addressing these problems, Campbell et al. [6] developed a time- and resource-economical self-report instrument: The Epistemic Trust, Mistrust and Credulity–Questionnaire (ETMCQ). The ETMCQ consists of 18 items, which can be grouped into three epistemic stances: The functional stance of trust, and the two maladaptive stances of mistrust and credulity. In the initial ETMCQ validation studies, the final 15-item model, where 3 low-loading items were removed, was confirmed with good model fit in the confirmatory factor analysis (CFA; [6]). Moreover, the ETMCQ had an acceptable internal consistency, as well as test-retest reliability [6]. Additionally, recent validations of the ETMCQ in other languages [13–15] have generally supported the three-factor structure. However, Liotti et al. [13] emphasized the need for further investigations of the factorial validity. Additionally, all three studies have excluded one or three items in their final versions. Subsequently, the validity of the ETMCQ requires further investigation, particularly in relation to its associations with psychopathology and other transdiagnostic constructs.

In addition, the ability of the ETMCQ to discriminate between more and less psychologically distressed individuals needs to be investigated. Following the hypothesis that ET may be modifiable in psychotherapy [2], application in clinical settings would enable to study ET as a treatment process variable.

## The present study

The main aim of this study is to investigate the factorial, convergent, and discriminant validity of the German version of the ETMCQ. Additionally, the discriminative power of the ETMCQ in a clinical and a non-clinical sample is examined. The following hypotheses are tested:

1. The factor structure with three correlating latent factors trust, mistrust, and credulity can be replicated in a German sample.

2. The subscales of the ETMCQ (trust, mistrust, credulity) can differentiate between participants of a clinical and a non-clinical sample.

3. Discriminant and convergent validity are examined in relation to early childhood trauma, mentalizing ability, and Criterion A and B of the AMPD. The three subscales of the ETMCQ (trust, mistrust, credulity) show different correlations with the related constructs:

   a. Higher scores in trust are positively related to higher mentalizing ability in relation to self and others.

b. Trust is negatively related to early childhood trauma, higher expression of maladaptive personality traits, and low personality functioning.

c. Mistrust and credulity are positively associated with early childhood trauma, higher expression of maladaptive personality traits, and low personality functioning.

d. Mistrust and credulity are negatively associated with mentalizing ability in relation to the self and others.

## Methods

### Study design

The cross-sectional study is part of a larger research project titled: "Transdiagnostic assessment of psychopathology and resilience: Psychometric evaluation of the German version of the Defense Mechanism Rating Scale (DMRS-SR-30) and relationship with personality functioning and associated constructs". The project was preregistered with PsychArchives [29], a repository for psychological science by the Leibniz Institute for Psychology (ZPID). The study was approved by the Ethics Committee of the Faculty of Behavioral and Cultural Studies of the University of Heidelberg, Germany (registration code: AZ Tau 2022 1/2).

### Participants and procedures

Detailed information about the recruitment of participants, sample characteristics and study procedures are reported in Volkert et al. [30]. Two independent samples were recruited. The data collection was conducted from 04/01/2022 to 04/20/2022 for Sample 1 and from 04/01/2022 to 07/14/2022 for Sample 2. Sample 1 ($N$ = 584) was recruited online using the German panel provider Respondi and funded by PsychLab (service provider of the Leibniz Institute for Psychology). The initial target sample size ($n$ = 490) was based on recommendations to obtain error-free correlation estimates by Kretzschmar and Gignac [31]. All participants provided written informed consent and completed online self-report questionnaires in a forced-choice format. Participants in sample 1 were aged between 25 and 74 ($M$ = 50.9, $SD$ = 13.7) and 367 identified as females (63%), 216 as males, and one participant as diverse. Within this first sample, 19% ($n$ = 112) of individuals had a diagnosis of at least one mental disorder over the past 12 months.

Sample 2 ($N$ = 60) was recruited independently from PsychLab. This group of 60 participants was divided into two subgroups: a clinical group ($n$ = 30) and a non-clinical group ($n$ = 30). In order to qualify for inclusion in the clinical subgroup, participants were required to have received a diagnosis for a mental disorder in accordance with either the ICD-10 or the DSM-5, and to be undergoing out- or inpatient psychiatric/ psychotherapeutic treatment at the time of data collection. The clinical subgroup was specifically recruited from two psychiatric inpatient facilities (Psychiatric Center in Wiesloch; General Psychiatric Hospital of the University of Heidelberg, Germany). In addition, the Heidelberg University Hospital website, mailing lists, and snowball-sampling were used to recruit participants for both subgroups. The participants in sample 2 provided written informed consent and completed the same online self-report questionnaire as participants in sample 1. Additionally, they completed a semi-structured interview, which is not part of this paper. The mean age in sample 2 was 30.8 ($SD$ = 13.6) and gender distribution was 39 females (65%), 19 males (32%), and two diverse participants (3%). The clinical subgroup predominantly suffered from depression (57%, $n$ = 17) and anxiety (33%, $n$ = 10). Of the 30 participants of the clinical subgroup, 20% ($n$ = 6)

of participants were psychiatric inpatients, while the remaining 24 participants received outpatient psychotherapy at the time of the study.

## Measures

**Sociodemographic measures.** The sociodemographic variables age, gender, highest level of education and occupational status were assessed for the sample description and associations with the three ETMCQ subscales. In addition, participants were asked whether they had a mental disorder within the last 12 months, and the general symptom severity was assessed by means of the Patient Health Questionnaire– 2 (PHQ-2) [32] for depressive symptoms and the Generalized Anxiety Disorder– 7 Scale (GAD-7) [33] for anxiety symptoms.

**ETMCQ.** The ETMCQ [6] includes a total of 15 items and was translated into German by Nolte et al. ([34]; translated items in S1 Table). The ETMCQ consists of three subscales, namely trust ("I find it easier to trust and take in information when it comes from someone who knows me well."), mistrust ("I often feel that others do not understand what I want and need."), and credulity ("Various people have told me that I am too easily influenced by others."). The five items per subscale are rated on a 7-point Likert scale ranging from 1 (*strongly disagree*) to 7 (*strongly agree*). For the test evaluation, the mean values were calculated for the items of the three subscales. McDonald's omega and Cronbach's alpha were in good to very good range for the trust ($\omega = .82$, $\alpha = .78$) and credulity subscales, for either the 15-item version ($\omega = .86$, $\alpha = .80$) and the 12-item version ($\omega = .87$; $\alpha = .79$). For the mistrust subscale, internal consistency was in the acceptable to good range ($\omega = .69$, $\alpha = .60$ / 12-item version: $\omega = .64$; $\alpha = .63$).

**Certainty About Mental States Questionnaire.** (CAMSQ; [35]) was used to assess the perceived certainty of mentalizing ability in relation to self and others. The CAMSQ consists of 20 items, of which 10 items relate to the interpretation and of one's own mental states (e.g. item 16: "I know the reasons for my behavior.") and the other 10 items relate to the interpretation of mental states in others (e.g. item 3: "I know how other people will react to something."). The items are each rated on a 7-point Likert scale ranging from 1 (*never*) to 7 (*always*). For the evaluation of the two subscales self-certainty and other-certainty, the mean values of the item responses were calculated. The discrepancy value between the two subscales was not part of the analyses. Internal consistency was very high for both subscales ($\omega = .92$; $\alpha = .91$).

**Personality Inventory for DSM-5 –Brief Form Plus Modified.** (PID5BF+M; [36]). To assess maladaptive personality traits, we used the PID5BF+M, the modified German short form of the Personality Inventory for DSM-5 [37]. This 36-item questionnaire in self-report format captures 18 trait facets reflecting the six domains of antagonism, psychoticism, disinhibition, negative affectivity, detachment, and anankastia. Item responses are assessed on a 4-point Likert scale ranging from 0 (*strongly disagree*) to 3 (*strongly agree*). Higher scores indicate more pronounced maladaptive personality traits. Internal consistency was in the good to very good range for all six domains in the present study ($\omega = .81$ - $.87$; $\alpha = .77$ - $.84$). For the evaluation of the six domains, the mean values of the respective item responses were calculated.

**Level of Personality Functioning Scale–Brief Form 2.0.** (LPFS-BF 2.0; [38]). In the AMPD impairments in personality functioning level are described with two core features, limitations in self-functioning (self) and interpersonal problems (other). The LPFS-BF 2.0 is the revised German short version of the Level of Personality Functioning Scale [39]. The two higher-level domains are captured with 12 items (self-subscale: e.g. item 1: "I often do not know who I really am."; other-subscale: e.g. item 10: "My relationships and friendships never last long."). Agreement with the statements of the items is assessed on a 4-point Likert scale

ranging from 1 (*not at all true or often untrue*) to 4 (*very true or often true*). The sum scores for the two subscales and the total score were used for further analyses in this study. Reliability proved to be excellent for the subscale self ($\omega$ = .91, $\alpha$ = .90). For the subscale other, McDonald's omega (.78) and Cronbach's alpha (.75) were in the good range. The reliability of the total scale is in the high to excellent range ($\omega$ = .90, $\alpha$ = .89).

**Childhood Trauma Questionnaire.** (CTQ; [40]). Experience of early childhood trauma was measured retrospectively using the German version of the CTQ short version [38]. The 28 self-report items capture five subscales of sexual, physical, and emotional abuse, as well as physical and emotional neglect. The items are rated on a 5-point Likert scale ranging from 1 (*not at all*) to 5 (*very often*) and higher scores indicate more severe neglect or abuse. The reliability of the physical neglect subscale proved to be the lowest in this study, but still in the good range ($\omega$ = .75, $\alpha$ = .71). The reliability of the other subscales and the total score were in the excellent range ($\omega$ = .88 - .96, $\alpha$ = .85 - .95). To evaluate the individual scales, the sum scores of the item responses were calculated; for the total score, the mean value was calculated.

## Statistical analysis

All statistical analyses were performed in RStudio [41] using R version 4.2.2 [42]. A description of the data cleaning performed prior to the statistical analyses can be found in the preregistration [29]. In the self-report sample 622 participants completed the study and passed the attention check item. After that, 19 participants from sample 1 were excluded based on the relative speed index [43]. Finally, the long string method was used to detect careless respondents [44], which resulted in the exclusion of 19 additional participants in sample 1 (*N* = 584). More details can be found in Volkert et al. [30].

**Associations with demographic features.** Exploratorily, we analyzed associations between sociodemographic variables, symptom severity and the three ETMCQ subscales with the data from self-report sample (*N* = 584). Due to the non-normal distribution of the data non-parametric Wilcoxon rank tests for two independent samples, Spearman correlations, and analysis of variances (ANOVAs) were conducted. Effect sizes were calculated where applicable.

**CFA.** CFA was performed on the data from sample 1 using the R package lavaan [45]. Testing for multivariate and univariate normal distribution using R package MVN [46] indicated deviations from normal distribution in the data (Mardia's test skewness = 70.44, $p <$ .001, kurtosis = 3.87, $p <$ .001; Shapiro Wilk test: $p <$ .001 - .008 for all three subscales of the ETMCQ). To identify multivariate outliers, the Mahalanobis distance (*D*) was calculated. This identified seven outliers (*D* = 14.91–25.49, $p <$ .001). However, the corresponding cases were not excluded for consistency reasons, as they showed an overall coherent response behavior. Due to these prerequisite violations, the robust estimation method Maximum Likelihood Robust (MLR) was used. To evaluate global goodness of fit of the model, various fit indices were considered in addition to the $\chi^2$-test statistic: The comparative fit index (CFI) and Tucker-Lewis index (TLI), with values above .90 indicating an acceptable model fit and above .95 indicating a good model fit between data and model [47]. Overall measures of model variances were calculated as the standardized root mean residual (SRMR; good model fit: < .08) and the root mean square error of approximation (RMSEA) with 90% confidence interval (CI) and *p* value, with nonsignificant values below .06 indicating a good model fit. First, four different models were investigated as part of the hypothesis testing: Two original ETMCQ model versions with 15 items, the first without additional model specifications, and the second with the correlated residuals adopted from Campbell et al. [6]. In the third and fourth model, items 3, 6 and 14 with the lowest factor loadings were removed and a 12-item version of the ETMCQ

without and with correlated residuals was tested (S2 Table). Subsequently, as a result of the high correlation between the latent factors mistrust and credulity, we calculated two further models with 2-factors and 12 items.

***t*-Tests.** To examine the differentiation between clinical and non-clinical samples (hypothesis 2), data from sample 2 was analyzed. Considering that the *t*-test proves to be relatively robust to the violation of the normal distribution assumption for a sample size $n \geq 30$ [48] and variance equality was given, two-tailed, unpaired two-sample *t*-tests were performed. The significance level was set at $p < .05$.

**Correlations.** Spearman correlations were calculated for the associations of the ETMCQ subscales with criterion A and B of the AMPD, as well as the severity of early childhood trauma, using the data from sample 1 (hypotheses H3b and H3c). The significance level of the inferential statistical test was set at $p < .05$ and was corrected by using the Bonferroni method for multiple testing. For the interpretation of the Spearman coefficient, $r = .10$ was considered a small, $r = .30$ a moderate, and $r = .50$ a strong correlation [49].

**Structural Equation Modeling (SEM).** The hypothesized relationship between the three subscales of the ETMCQ and mentalizing was analyzed using structural equation modeling (Hypotheses H3a and H3d). The SEM consists of five latent factors: the three subscales of the ETMCQ (trust, mistrust, and credulity) and the two dimensions of mentalizing ability of the CAMSQ (self-certainty and other-certainty). The factors were operationalized by the corresponding items and were allowed to correlate with each other. The prerequisite test for the two CAMSQ subscales revealed a deviation from a normal distribution (Mardia's test: skewness = 44.73, $p < .001$, kurtosis = 4.08, $p < .001$; Shapiro Wilk test: $p < .001$, $p = .020$ for the self-certainty and other-certainty subscales, respectively). The univariate skew values for each indicator variable were between -1 and +1 (self-certainty: - 0.59; other-certainty: -0.23; trust: -0.67; mistrust: -0.16; credulity: 0.46). Using Mahalanobis distance (*D*), seven outliers were identified (*D* = 20.80–33.56, $p < .001$), but were not excluded due to content conclusive response. Because of these prerequisite violations, the MLR estimation method was used for the CFA of the CAMSQ. The CFA showed acceptable model fit of the data indicated by the fit indices, $\chi^2 = 477.88$, $df = 169$, $p < .001$; CFI = .94; TLI = .93; RMSEA = .06, 90% CI [.06 - .07], $p = .030$; SRMR = .04. To test for multicollinearity among the five latent factors, multiple regressions were performed, and variance inflation factors (VIF) were calculated. No multicollinearity was found ($R^2 < .50$; VIF < 2). The same test statistics and fit indices were used to assess model goodness of fit as for the CFA on ETMCQ.

# Results

## Descriptive results

Table 1 shows the descriptive statistics (means, standard deviations, ranges), as well as the reliabilities of all scales and subscales. In sample 1, mentalizing ability (CAMSQ) and maladaptive personality traits (PID5BF+M) showed similar levels as in the representative validation study of the CAMSQ [35], respectively compared to preliminary norm values for the German general population [50; supplementary material]. Impaired personality functioning level (LPFS-BF total score) was above thresholds recommended by Spitzer et al. [38].

In terms of early childhood traumatization, the overall burden was similar when compared to the samples in Campbell et al. [6]. However, compared to Campbell et al. [6] and a representative German population sample [51], our sample 1 showed a higher percentage of reported moderate to extreme physical neglect (22.9% with scale score $\geq 10$ [51, 52]): 20.5% reported moderate to extreme emotional abuse (scale score $\geq 13$;); 16.1% reported moderate to extreme

**Table 1. Descriptive statistics and internal consistencies in sample 1 ($N$ = 584).**

| Scale | Range | $M$ | $SD$ | ω | α |
|---|---|---|---|---|---|
| ETMCQ (15-/12-item version) | | | | | |
| Trust | 1–7 | 4.9 | 1.0 | .82 | .78 |
| Mistrust | 1.6–6.8 / 1–7 | 4.2 / 4.2 | 0.9 / 1.2 | .69 / .64 | .60 / .63 |
| Credulity | 1–7 | 3.2 / 3.1 | 1.2 / 1.3 | .86 / .87 | .80 / .79 |
| CAMSQ | | | | | |
| Self-certainty | 1.2–7 | 5.2 | 1.0 | .92 | .91 |
| Other-certainty | 1.1–7 | 4.6 | 0.9 | .92 | .91 |
| CTQ | | | | | |
| Physical neglect | 5–25 | 7.9 | 3.4 | .75 | .71 |
| Physical abuse | 5–25 | 6.9 | 3.8 | .95 | .92 |
| Emotional neglect | 5–25 | 11.3 | 5.5 | .94 | .92 |
| Emotional abuse | 5–25 | 8.8 | 4.8 | .92 | .89 |
| Sexual abuse | 5–25 | 6.0 | 3.0 | .96 | .95 |
| Total | 5–21.8 | 8.2 | 3.3 | .88 | .85 |
| PID5BF+M | | | | | |
| Negative affectivity | 0–3 | 1.1 | 0.6 | .86 | .82 |
| Detachment | 0–3 | 0.9 | 0.6 | .82 | .78 |
| Antagonism | 0–2.8 | 0.6 | 0.5 | .82 | .77 |
| Disinhibition | 0–2.7 | 0.8 | 0.5 | .81 | .77 |
| Anankastia | 0–3 | 1.2 | 0.7 | .87 | .84 |
| Psychoticism | 0–3 | 0.8 | 0.6 | .82 | .78 |
| LPFS-BF 2.0 | | | | | |
| Self | 6–24 | 10.9 | 4.6 | .91 | .90 |
| Other | 6–24 | 10.7 | 3.3 | .78 | .75 |
| Total | 12–48 | 21.5 | 7.3 | .90 | .89 |

*Note*. ETMCQ = Epistemic Trust Mistrust and Credulity Questionnaire. CAMSQ = Certainty About Mental States Questionnaire. CTQ = Childhood Trauma Questionnaire. PID5BF+M = Personality Inventory for DSM-5, Brief Form Plus Modified. LPFS-BF 2.0 = Level of Personality Functioning Scale–Brief Form 2.0. $M$ = Mean. $SD$ = Standard deviation. ω = McDonald's omega total. α = Cronbach's alpha.

physical abuse (scale score ≥ 10); 11.8% reported moderate to extreme sexual abuse (scale score ≥ 8); and 26.3% reported moderate to extreme emotional neglect (scale score ≥ 15).

## Associations with demographic features

The nonparametric Wilcoxon rank test for two independent samples showed that women and men differed significantly in the expression of the subscale trust, but the effect was small ($M_{Women}$ = 5.0, $SD$ = 1.0; $M_{Men}$ = 4.8, $SD$ = 1.0; $W$ = 44385, $p$ = .015; $r$ = .10). For mistrust and credulity, there were no statistically significant group differences with respect to gender (mistrust: $M_{Women}$ = 4.2, $SD$ = 0.9; $M_{Men}$ = 4.2, $SD$ = 0.9; $W$ = 40201, $p$ = .772; credulity: $M_{Women}$ = 3.2, $SD$ = 1.2; $M_{Men}$ = 3.1, $SD$ = 1.1; $W$ = 40475, $p$ = .671). For trust, there was no significant difference between participants with or without a diagnosed mental disorder in the past 12 months ($M_{yes}$ = 5.0, $SD$ = 1.0; $M_{no}$ = 4.9, $SD$ = 1.0; $W$ = 27769, $p$ = .403). For mistrust ($M_{yes}$ = 4.5, $SD$ = 1.0; $M_{no}$ = 4.1, $SD$ = 0.9; $W$ = 31950, $p$ < .001; $r$ = .14) and credulity ($M_{yes}$ = 3.8, $SD$ = 1.3; $M_{no}$ = 3.1, $SD$ = 1.1; $W$ = 34854; $p$ < .001; $r$ = .22) significantly higher mean scores were found in the presence of a mental disorder, but with small effect sizes. Regarding the relationship between the three subscales of the ETMCQ and age Spearman correlation coefficients showed that trust ($r$ = -.21, $p$ < .001) and credulity ($r$ = -.12, $p$ = .004) correlated low but significantly

with age. Mistrust, on the other hand, showed no correlation with age ($r = .003$, $p = .951$). Moderate significant correlations with general symptom severity were found for mistrust (PHQ-2: $r = .34$, p $< .001$; GAD-7: $r = .33$, $p < .001$) and credulity (PHQ-2: $r = .34$, $p < .001$; GAD-7: $r = .36$, $p < .001$). Trust did not show statistically significant correlations with psychological distress (PHQ-2: $r = -.03$, $p = .446$; GAD-7: $r = .05$, $p = .263$). Mean differences in ETMCQ subscales related to education were examined using ANOVAs. Trust, $F(7, 576) = 1.071$, $p = .380$ and credulity, $F(7, 576) = 1.501$; $p = .164$, showed no significant group differences by educational degree. For mistrust, on the other hand, there was a significant difference with respect to education, but with a small effect size, $F(7, 576) = 2.566$; $p = .013$, $\eta2 = 0.03$. Post hoc pairwise group comparisons showed significant group differences in mistrust between (applied) university degree and completed apprenticeship/ training ($p = .017$; corrected for multiple comparisons using the Bonferroni method).

## ETMCQ factor structure

Both 15-item models and the 12-item model without correlated residuals showed an insufficient fit to the data (models 1–3; Table 2). In contrast, as shown in Table 2, model 4 with 12 items and correlated residuals had a good model fit. As shown in Fig 1, the latent factors trust and mistrust ($r = -.152$, $p = .034$) were negatively correlated at a 5% significance level, whereas trust and credulity showed no significant correlations ($r = .078$, $p = .224$). However, in model 4 the latent factors mistrust and credulity ($r = .824$, $p < .001$) were highly and significantly correlated with each other. Parameter estimates had no negative variances and all items loaded significantly on their corresponding factor (.522 - .811; Fig 1). The subsequently tested models 5 and 6 with two factors and 12-items showed insufficient model fit. Likelihood ratio tests showed a significant better model fit of model 4 against model 2 ($\chi^2_{diff} = 130.66$, $df_{diff} = 33$, $p > .000$) and a significant better model fit of model 4 against model 6 ($\chi^2_{diff} = 39.46$, $df_{diff} = 2$, $p > .000$). The standardized residual correlations for models 1, 2 and 4 can be seen in S4 Table.

## Discriminative power: Clinical vs. Non-clinical sample

In descriptive mean comparison, the clinical subsample ($n = 30$) showed higher scores on all three ETMCQ subscales than the nonclinical sample ($n = 30$), but $t$-tests revealed no significant group differences (Table 3). Hypothesis 2 could not be confirmed.

**Table 2. Results of confirmatory factor analysis, self-report sample 1 ($N = 584$).**

| Model | χ2 | CFI | TLI | RMSEA (90%—CI) | SRMR | AIC | BIC (sample size adjusted BIC) |
|---|---|---|---|---|---|---|---|
| Model 1: 15 items, 3-factors | $\chi^2(87, n = 584) = 460.114$, $p < .001$ | .82 | .78 | .092 (.079 - .093), $p < .001$ | .074 | 29567.850 | 29777.605 (29625.223) |
| Model 2: 15 items, 3-factors, correlated residuals | $\chi^2(82, n = 584) = 248.738$, $p < .001$ | .92 | .90 | .064 (.055 - .73), $p = .028$ | .067 | 29334.430 | 29566.035 (29397.779) |
| Model 3: 12 items, 3-factors | $\chi^2(51, n = 584) = 246.218$, $p < .001$ | .87 | .85 | .088 (.077 - .099), $p < .001$ | .052 | 23615.450 | 23785.876 (23662.065) |
| Model 4: 12 items, 3-factors, correlated residuals | $\chi^2(49, n = 584) = 119.912$, $p < .001$ | .96 | .94 | .054 (.042 - .067), $p = .497$ | .043 | 23471.046 | 23650.212 (23520.052) |
| Model 5: 12 items, 2-factors | $\chi^2(53, n = 584) = 294.303$, $p < .001$ | .86 | .82 | .096 (.085 - .107), $p < .001$ | .064 | 23667.861 | 23829.547 (23712.086) |
| Model 6: 12 items, 2-factors, correlated residuals | $\chi^2(51, n = 584) = 158.669$, $p < .001$ | .94 | .92 | .066 (.054 - .077), $p = .014$ | .055 | 23512.944 | 23683.370 (23559.559) |

*Note.* CFI = Comparative Fit Index. TLI = Tucker-Lewis Index. RMSEA = Root Mean Square Error of Approximation. SRMR = Standardized root Mean Square Residual. AIC = Akaike-Information-Criterion. BIC = Bayesian-Information-Criterion.

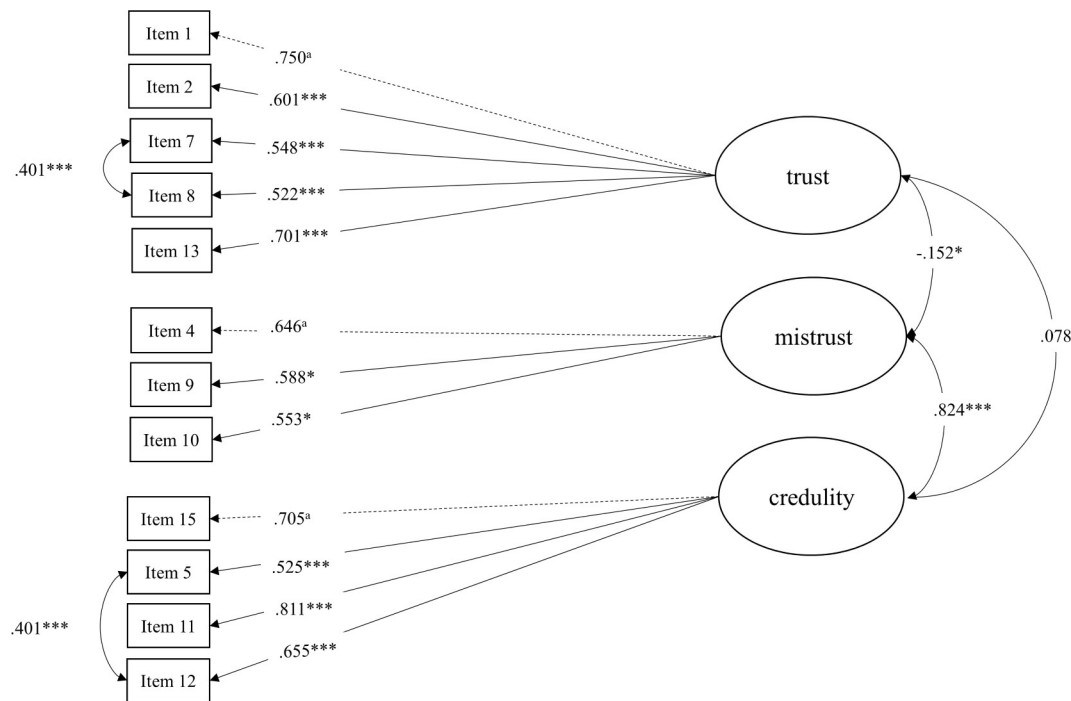

**Fig 1. CFA model 4 with factor loadings, self-report sample 1 ($N$ = 584).** Model 4 with 12 items and correlated residuals. All loadings are standardized. Measured variables are indicated by squares and latent factors by ovals. Note the item numbering is retained from the 15-item scale of the English version [6]. [a]Factor loading fixed to 1. $*p$ < .05. $***p$ < .001.

## Discriminant and convergent validity

Table 4 displays the Spearman correlation coefficients between the ETMCQ subscales and related constructs. As hypothesized, significant negative correlations were found for trust with childhood trauma in general and also with emotional neglect, physical neglect, and small, but significant, with physical abuse. There were no significant correlations between trust and emotional abuse, as well as sexual abuse. The predicted direction (hypothesis H3b) of the correlations between trust and maladaptive personality traits was shown only for detachment. There were no significant correlations found for trust with the maladaptive traits antagonism, disinhibition, anankastia, and psychoticism. Against our expectations, there was a significant positive correlation between trust and negative affectivity. Similarly, the presumed negative associations for trust with personality functioning did not appear for the total LPFS-BF score or for the self-subscale. However, and as expected, a small significant negative correlation was found for trust and personality functioning in relation to others. For mistrust and credulity,

**Table 3. ETMCQ scores of clinical and non-clinical participants, sample 2 ($N$ = 60).**

| ETMCQ | Clinical ($n$ = 30)<br>$M$ ($SD$) | Non-clinical ($n$ = 30)<br>$M$ ($SD$) | $t$-test | $p$ (95%—CI) |
|---|---|---|---|---|
| Trust | 5.6 (0.96) | 5.45 (0.83) | $t(58) = 0.64$ | .53 (-.32 - .61) |
| Mistrust | 3.99 (1.35) | 3.84 (1.15) | $t(58) = 0.45$ | .66 (-.51 - .79) |
| Credulity | 2.98 (1.48) | 2.7 (1.03) | $t(58) = 0.84$ | .41 (-.38 - .93) |

*Note*. ETMCQ = Epistemic Trust, Mistrust and Credulity–Questionnaire. $M$ = Mean. $SD$ = Standard deviation. CI = Confidence interval.

**Table 4. Spearman correlations between the ETMCQ subscales and related constructs, sample 1 ($N$ = 584).**

|  | Trust | Mistrust | Credulity |
|---|---|---|---|
| CAMSQ |  |  |  |
| Self-certainty | .22*** | -.16*** | -.25*** |
| Other-certainty | .18*** | .02 | -.15*** |
| CTQ |  |  |  |
| Physical neglect | -.15*** | .27*** | .24*** |
| Emotional abuse | -.05 | .33*** | .37*** |
| Emotional neglect | -.22*** | .30*** | .30*** |
| Physical abuse | -.11*** | .19*** | .24*** |
| Sexual abuse | .01 | .16*** | .17*** |
| Total | -.15*** | .35*** | .37*** |
| PID5BF+M |  |  |  |
| Negative affectivity | .18*** | .38*** | .45*** |
| Detachment | -.25*** | .40*** | .31*** |
| Antagonism | .04 | .22*** | .19*** |
| Disinhibition | .05 | .35*** | .44*** |
| Anankastia | .07 | .29*** | .24*** |
| Psychoticism | .01 | .41*** | .37*** |
| LPFS-BF 2.0 |  |  |  |
| Self | .03 | .39*** | .47*** |
| Other | -.12** | .40*** | .39*** |
| Total | -.04 | .44*** | .48*** |

*Note*. ETMCQ = Epistemic Trust, Mistrust and Credulity Questionnaire. CAMSQ = Certainty About Mental States Questionnaire. CTQ = Childhood Trauma Questionnaire. PID5BF+M = Personality Inventory for DSM-5, Brief Form Plus Modified. LPFS-BF 2.0 = Level of Personality Functioning Scale–Brief Form 2.0.

**p < .01

***p < .001.

overall, slightly stronger correlations were found in the expected direction. Early childhood trauma showed small to moderate correlations with mistrust and credulity. Mistrust and credulity correlated positively in small to moderate strength with all six dimensions of maladaptive personality traits (PID5BF+M). Credulity showed the strongest correlations with negative affectivity and disinhibition, and mistrust showed the strongest correlations with psychoticism, detachment, and negative affectivity. In addition, and as expected, positive correlations of moderate strength were found between personality functioning impairments and mistrust and credulity.

**Mentalizing and epistemic stance.** The SEM to assess the associations between the ETMCQ and the CAMSQ showed an acceptable model fit to the data of sample 1, $\chi^2$ = 1032.94, $df$ = 452, $p < .001$; CFI = .92; TLI = .91; RMSEA = .051, 90% CI [.047, .055], $p$ = .929; SRMR = .054. The model with the respective standardized path coefficients indicating the strength of the relationships between the latent factors of the ETMCQ scales and the latent factors of the CAMSQ is shown in Fig 2. As predicted, significant positive associations were found between trust and self-certainty and other-certainty, respectively (β = .26, $p < .001$; β = .20, $p$ = .002). The relationships between mistrust and credulity and the two CAMSQ scales showed mixed results. As hypothesized, mistrust and self-certainty were significantly negatively related (β = —.26, $p$ = .001). However, there was no significant relationship between mistrust and other-certainty (β = .03, $p$ = .671). In line with the hypothesis (3d), the relationships between credulity and self-certainty (β = —.28, $p < .001$) and other-certainty (β = -.14, $p$ =

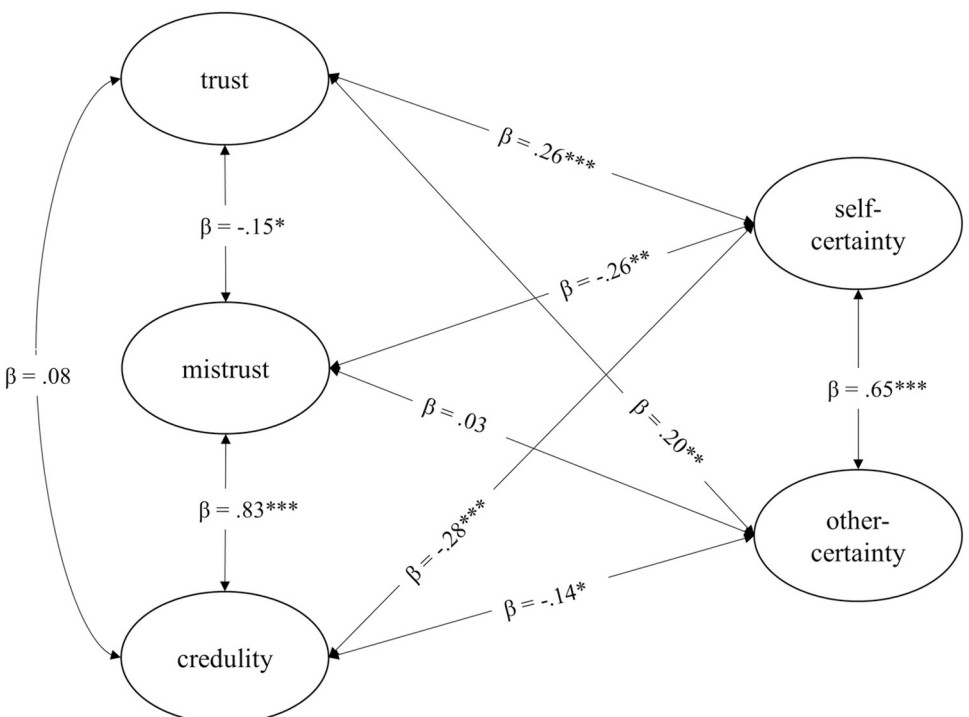

**Fig 2. SEM: Associations between mentalizing and epistemic stance.** All path coefficients are standardized. Latent factors for epistemic stance: Trust, mistrust, and credulity. Latent factors for mentalizing: Self-certainty and other-certainty. *** $p < .001$; ** $p < .01$; * $p < .05$.

.020) proved to be significantly negative. The standardized path coefficients between the trust and mistrust factors of the ETMCQ indicated a significant negative relationship ($\beta = -.15$, $p = .034$). Trust and credulity did not show a significant relationship ($\beta = .08$, $p = .203$). Mistrust and credulity were significantly positively associated ($\beta = .83$, $p < .001$). The two latent factors of the CAMSQ were significantly positive related ($\beta = .65$, $p < .001$). The parameter estimates had no negative variances and all items loaded significantly on their corresponding factor (.535 - .777; see all factor loadings in S3 Table).

## Discussion

The aim of this study was to investigate the factorial validity of the German version of the ETMCQ [34] and to extend knowledge about convergent and discriminant validity.

In summary, our results yield further empirical support for the three-factor structure of the ETMCQ and support the assumption that trust, mistrust, and credulity are distinct strategies of dealing with socially transmitted information. However, the factor structure was only replicated with a sufficient model fit to the data using a shortened 12-item version in our German sample. Our findings align with those of previous validation studies, which have all identified an optimal model fit only for shortened versions of the ETMCQ [13–15]. Compared to Campbell et al. [6], the CFA results showed deviations in the correlations among the latent factors. Especially, the substantially weaker negative correlation between trust and mistrust and the considerably stronger positive correlation between mistrust and credulity are notable. The high correlation between these two dimensions can be supported by the assumption that without ET, rapid shifts between credulity and mistrust could occur [12]. On the other hand, it is reasonable to conclude that mistrust and credulity may not be separate factors. This suggests

that the ETMCQ may be interpreted as capturing a two-dimensional construct. The subscales of mistrust and credulity would represent two manifestations of maladaptive epistemic stance, and the trust subscale would represent the adaptive stance. However, subsequent testing of the two-factor models (models 5 and 6) to verify this assumption revealed a poorer fit to the data than the three-factor model (model 4). In line with Campbell et al. [6] and Liotti et al. [13], we found no significant relationship between trust and credulity, whereas Greiner et al. [14] and Asgarizadeh & Ghanbari [15] found small positive association between trust and credulity. Yet, the differing correlations between credulity and trust, and trust and mistrust are theory-supporting indications that mistrust and credulity are strongly related but distinct factors. However, and in particular with regard to the high correlation between mistrust and credulity, the potential impact of reducing the mistrust scale to only three items should be considered, despite the fact that the common recommendation of at least three indicators per factor in CFA in combination with an adequate sample size has been met [53, 54]. In summary, invariance testing and further investigation of the ETMCQ factor structure and potential influence of cultural and linguistic factors on the discrepancies observed in the separate validation studies is needed to improve the psychometric properties of this promising measure.

Contrary to hypothesis 2, the ETMCQ did not distinguish between a clinical and a non-clinical sample. Descriptively, the clinical sample showed higher scores in mistrust, credulity and, against expectations, also higher trust scores than participants in the non-clinical sample. However, these differences were not statistically significant. The relatively small sample size is the primary reason for this outcome, as only large effects could have been detected with the given power. In addition, another problem may arise from the composition of the clinical sample. Most participants in the clinical sample received long-term outpatient psychotherapy and only six participants were in inpatient treatment at the time of the assessment. Conceivably, long-term outpatient psychotherapy could have already had a positive effect on ET. As our study yielded inconclusive results regarding the differential strength of the ETMCQ, future research should endeavor to replicate this investigation with a larger sample size.

Overall, the directions of the correlations assumed in hypothesis 3 for the subscales mistrust and credulity could be broadly confirmed. The expected correlations between trust and the examined constructs were only partially shown. As predicted, early childhood trauma is positively associated with both maladaptive epistemic stances, mistrust and credulity. These findings align with those reported by Campbell et al. [6]. Individuals, who report more neglect and abuse, exhibit stronger epistemic mistrust and stronger credulity. This is consistent with the theoretical assumption that harmful childhood experiences can lead to the development of mistrust and limit the ability to adequately identify trustworthy sources [4]. These findings are also supported by studies examining associations between harmful childhood experiences and social cognition [55] and biases in the processing of social information [56]. For example, children with experiences of physical abuse have a higher sensitivity to the recognition of anger, an attention bias in relation to frightening social information, and a more hostile attribution style in neutral or ambiguous situations [56].

According to hypothesis (3b), trust is negatively associated with physical and emotional neglect. Trust is also weakly negatively associated with physical abuse, but there is no relevant association between trust and emotional and sexual abuse. This is in line with Campbell et al. [6] and Benzi et al. [17], who also found no negative association between trust and abuse. Following considerations of Campbell et al. [6], the lack of a link between trust and abuse may be seen as further evidence of differential effects on the emergence of ET depending on the nature of the damaging childhood experiences. Further investigations, including larger clinical samples, should examine ET as a core element of social learning to different neurobiological effects of abuse and neglect [57].

As hypothesized, mistrust and credulity were both positively associated with all six dimensions of maladaptive personality traits. Mistrust showed the strongest associations with detachment and psychoticism. The domain detachment, composed of the facets withdrawal, anhedonia, and intimacy avoidance, describes dysfunctional traits [34, 46] that could be the result of epistemic mistrust. Similar considerations could be made regarding the domain psychoticism (unusual beliefs and experiences; eccentricity; perceptual dysregulation; [36, 50]). The general expectation of not receiving relevant and reliable information from other people and the subsequent feeling of isolation may lead to difficulties in functioning in the social world. These difficulties may be reflected in the higher expressions of maladaptive personality traits. Credulity showed the strongest correlations with negative affectivity and disinhibition. The domain negative affectivity is composed of the personality facets emotional lability, anxiety, and separation anxiety; the domain disinhibition is described by the facets distractibility, impulsivity, and irresponsibility [36, 50]. One possible interpretation of the correlations found is that credulity may be associated with the development of a diffuse self-image [5] and therefore leads to personality traits associated with instability.

In summary, the different emphases in the associations of maladaptive personality traits with mistrust and credulity confirm the assumption that they are independent, learned strategies in dealing with social communication. Furthermore, the concept of different epistemic stances offers an explanatory approach to the social learning processes between adverse childhood experiences and maladaptive personality traits, which is also supported by the findings of Back et al. [58] on maladaptive traits and childhood trauma.

For the trust subscale, the expected negative associations only appeared with detachment. Contrary to our hypothesis, trust was not associated with antagonism, disinhibition, anankastia, or psychoticism. Also contrary to our expectations, trust and negative affectivity showed weak positive associations. One possible explanation for this positive association may be the disputed informative value of the negative affectivity trait domain [59]. This trait domain explains a significant portion of the variance in personality pathology and should be used in conjunction with other trait domains to obtain relevant information [59]. Thus, as described for early childhood traumatization, trust shows more heterogeneous associations than mistrust and credulity. To validate the observed patterns of specific correlations between the subscales of the ETMCQ and criterion B of the AMPD for consistency, replications of this study using different samples are necessary. Comparative studies with other measures of maladaptive personality traits, such as the Personality Inventory for ICD-11 [60, 61] additionally support this purpose. Moreover, the expected negative associations between low personality functioning (criterion A of the AMPD) and trust were not found for the dimension self or overall severity. However, regarding the dimension other, participants with higher values in trust showed less impairment in personality functioning. Yet, this correlation was rather small.

The empirical findings relating to maladaptive traits and personality functioning could serve as indications that ET may be a neutral mechanism in the sense of a human default mode [6, 62]. As considered by Campbell et al. [6], higher levels of ET may not always act as an active resilience factor, and the benefits for social interactions may be exhausted beyond a certain level. However, this does not appear to apply uniformly to all constructs and subscales studied. Further examination of these phenomena in additional samples can provide insight into whether ET can be considered a resilience factor in specific areas or if the trust subscale does not fully capture the construct. The theoretical concept of ET as a transdiagnostic resilience factor [12], which coexists with default mode considerations, requires focused examination. To ensure objectivity, it is important to note the correlational nature of these results and their relatively small values.

In line with the hypothesis (3c), impairments in personality functioning were positively associated with mistrust and credulity. The findings support the theoretical assumptions about links between epistemic mistrust and credulity and personality functioning. Whether mistrust and credulity have causal effects on the development of personality functioning needs to be investigated in longitudinal studies.

The SEM for investigating the relationship between epistemic stance and mentalizing showed an acceptable overall model fit. However, the assumed relationships between the epistemic stances and mentalizing can only be partially confirmed. According to the hypothesis (3a), higher certainty about one's own mental states (self-certainty) and mental states of others (other-certainty) were associated with higher values in trust as a learned strategy in dealing with social knowledge transfer. As hypothesized (3d), mistrust and credulity were negatively related with self-certainty. These findings are consistent with theoretical considerations that difficulties in adequately assessing socially transmitted knowledge are associated with deficits in mentalizing ability [12] and are in line with the previous results of Campbell et al. [6].

Additionally, the relationship between credulity and other-certainty points in its predicted direction. Contrary to the hypothesis (3d), mistrust and certainty about mental states of others (other-certainty) were not associated in this study. Thus, the basic attitude of not receiving relevant, trustworthy, and generalizable information from others might be inconsistent with deficits in certainty about others' mental states. This cautious interpretation of the results is consistent with the unclear patterns of results regarding the maladaptivity of low other-certainty found by Müller et al. [35]. However, it needs to be considered that the correlative SEM design does not allow inference about causal relationships. Second, the CAMSQ does not capture mentalizing in all facets. Müller et al. [35] noted that the CAMSQ measures certainty about mental states rather than the accuracy of these attributions. To comprehensively understand the complex interactions between mentalizing and epistemic stance, future research should also investigate the relationship between the ETMCQ and the accuracy of mentalizing ability, as measured by performance tasks (e.g., Reading the Mind in the Eyes Test [63]; Movie for the assessment of social cognition [64]). Additionally, further analysis of the relationship between trust, mistrust, and credulity with hypermentalizing is needed. For this purpose, the discrepancy value between self- and other-certainty should be used since a high level of certainty about mental states of others has proven to be maladaptive only in relation to observed low self-certainty [35]. Nevertheless, the complexity of the interrelationships between epistemic stances and mentalizing becomes visible, and further exploration of these relationships will provide worthwhile insights into previously predominantly theoretical assumptions.

## Limitations

In addition to the limitations already mentioned, the following points should be considered. First, early childhood trauma was recorded retrospectively in self-report. When interpreting the results, it is important to keep in mind that prospective and retrospective measures of childhood trauma identify different groups of individuals, and the results found are not equivalent [65, 66]. Second, sample 1 reported more experiences of traumatic events compared to a representative sample of the German population [51]. One possible explanation for this could be the higher mean age in our sample, as older age is associated with more experiences of physical neglect [67]. Another explanation for more traumatic childhood experiences in sample 1 might be a self-selection bias, which has been reported in other studies [68]. Additionally, other constructs (mentalizing, personality functioning and traits, etc.) were also only assessed using self-report instruments. Third, the cultural affiliation of the participants was not recorded, which makes a possible influence of culture on the results impossible to investigate.

The cultural background may influence the social adaptivity of certain manifestations of ET, as well as mistrust and credulity, or the relationship between epistemic stance and mentalizing [69, 70]. Furthermore, our samples are from a western, educated, industrialized, rich, democratic (WEIRD) society, which needs to be considered when interpreting our findings [71]. Fourth, this is a cross-sectional study, which does not allow to draw any conclusions about potential causal relationships between the investigated variables.

## Implications and future directions

Important implications for future research and clinical work emerge from our findings. Longitudinal and cohort studies are needed to draw more substantiated conclusions about the potential causal role of epistemic stance in the emergence of psychopathology. Additionally, future multimethod approaches may provide more comprehensive insights into the construct ET and underpin the construct validity of the ETMCQ. Comparing the ETMCQ with experimental measurement methods, such as the Epistemic Trust Assessment [24], would allow for the investigation of situation-dependent aspects of ET, mistrust and credulity. Moreover, investigating epistemic stance as a cross-cultural construct, and whether culture-dependent profiles with varying degrees of trust, mistrust, and credulity exist, can be a useful future endeavor.

The development of the ETMCQ provides new opportunities to address further theoretical and clinical aspects. It is important to address existing criticism of the concept of ET, such as the imprecise use of the terms trust and mistrust, as well as the lack of classification of ET in the context of interdisciplinary trust research [72]. Therefore, future research should aim to clarify the differences and connections between trust concepts, including its relationship to interpersonal trust, as suggested by Botsford and Renneberg [73]. Additionally, in accordance with Fonagy et al.'s [4] introduction of ET, mistrust, and credulity to the discussion on a general factor of psychopathology and building upon the suggestions made by Wendt et al. [74], exploring the connections and potential integration of the ETMCQ into the Hierarchical Taxonomy of Psychopathology (HiTOP; [75]) could be a future research objective. Additionally, it is important to implement and evaluate therapeutic interventions that focus on building ET in patients with mental disorders. One potential avenue to pursue, is integrating the Mediational Intervention for Sensitising Caregivers [76, 77] into mentalization-based treatment with the goal of operationalizing the (re-) establishment of ET through psychotherapy. Another way could be to integrate the EMTCQ in discovery-oriented approaches to explore state-like and trait-like changes in trust, mistrust, and credulity through psychotherapy process [78]. The ETMCQ [6] could be used to empirically test interindividual profiles of epistemic stance. Furthermore, a reliable and valid measurement tool for epistemic stance is valuable in gaining insights into social communication mechanisms. Recent research has explored the association between epistemic stance and belief in conspiracy theories and found heterogenous results [79, 80]. This would enable the development of individually tailored public mental health interventions [81].

## Conclusions

This study reports results on the validation of the German version of the ETMCQ [11, 32] and is the first study investigating this novel measure of ET, mistrust, and credulity in a clinical and non-clinical sample. With the ETMCQ, a first, reliable, valid, and test-economic measurement instrument for epistemic stance is available. The found associations between mistrust and credulity with mentalizing abilities, early childhood trauma, maladaptive personality traits, and personality functioning largely reinforce the previous theoretical and empirical studies on

the role of epistemic stance in psychopathology. Therewith, our findings also support the notion that epistemic stance may provide a fruitful etiological starting point for a deeper understanding of the development of psychopathology. However, further studies of the ETMCQ, including investigation in larger, more heterogenous clinical groups and as a process variable in psychotherapeutic treatment, as well as translation into other languages and cultural adaptation will be needed.

## Supporting information

**S1 Table. German and English ETMCQ items.**
(DOCX)

**S2 Table. CFA model 2 factor loadings.**
(DOCX)

**S3 Table. SEM factor loadings.**
(DOCX)

**S4 Table. Standardized residual correlations for CFA models.**
(DOCX)

**S1 File. Sample 2 dataset.**
(CSV)

## Acknowledgments

We thank all participants in both studies for their engagement.

## Author Contributions

**Conceptualization:** Svenja Taubner, Max Zettl, Jana Volkert.

**Data curation:** Anna-Maria Weiland, Leonie C. Bartmann, Nina Frohn, Mirijam Luginsland.

**Formal analysis:** Anna-Maria Weiland, Leonie C. Bartmann, Nina Frohn, Mirijam Luginsland.

**Funding acquisition:** Max Zettl, Jana Volkert.

**Investigation:** Anna-Maria Weiland, Max Zettl, Leonie C. Bartmann, Nina Frohn, Mirijam Luginsland, Jana Volkert.

**Methodology:** Anna-Maria Weiland, Max Zettl, Jana Volkert.

**Project administration:** Max Zettl, Jana Volkert.

**Supervision:** Svenja Taubner, Max Zettl, Jana Volkert.

**Writing – original draft:** Anna-Maria Weiland.

**Writing – review & editing:** Anna-Maria Weiland, Svenja Taubner, Max Zettl, Leonie C. Bartmann, Nina Frohn, Mirijam Luginsland, Jana Volkert.

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
