## [Decision Letter · Decision Letter 0]

3 Nov 2023

PONE-D-23-22407Epistemic trust and associations with psychopathology: Validation of the German version of the Epistemic Trust, Mistrust and Credulity-Questionnaire (ETMCQ)PLOS ONE

Dear Dr. Weiland,

Thank you for submitting your manuscript to PLOS ONE. After careful consideration, we feel that it has merit but does not fully meet PLOS ONE’s publication criteria as it currently stands. Therefore, we invite you to submit a revised version of the manuscript that addresses the points raised during the review process.

We look forward to receiving your revised manuscript.

Kind regards,

Hanna Landenmark

Staff Editor

PLOS ONE

Journal Requirements:

**Additional Editor Comments:**

Please see the comments from two reviewers below. Please note especially that reviewer 2 queries whether the German translated questionnaire is available, and whether any studies noted as "forthcoming" have already performed a validation of this questionnaire. Please clearly outline in your response whether the translated questionnaire and the forthcoming studies are available, and how other researchers may access these. We note that some of the reviewer comments may be seen a little terse in tone in places, for which we apologise, but overall I hope that the comments are useful to you in order to revise the manuscript and strengthen the study.

Reviewers' comments:

Reviewer's Responses to Questions

**Comments to the Author**

1. Is the manuscript technically sound, and do the data support the conclusions?

Reviewer #1: Partly

Reviewer #2: Partly

2. Has the statistical analysis been performed appropriately and rigorously? 

Reviewer #1: No

Reviewer #2: Yes

3. Have the authors made all data underlying the findings in their manuscript fully available?

Reviewer #1: Yes

Reviewer #2: Yes

4. Is the manuscript presented in an intelligible fashion and written in standard English?

Reviewer #1: Yes

Reviewer #2: No

5. Review Comments to the Author

Reviewer #1: ID: PONE-D-23-22407

Title: Epistemic trust and associations with psychopathology: Validation of the German version of the Epistemic Trust, Mistrust and Credulity-Questionnaire (ETMCQ)

Thank you for providing a chance to review this manuscript.

Detailed information:

Introduction

Line 56-59, Page 3: What are the further conceptual groundwork proposed by Sperber et al. and Wilson and Sperber? This is what needs to be clarified.

Paragraph 1, Page 3-4: So, on what basis, does the authors focu on such a psychosocial phenomenon? Are there any relevant statistics to support such a psychological problem on a social level?

Line 80-86, Page 4: How strong is the association? In addition, the authors give many new concepts in this paragraph, and if the authors do not intend to elaborate in depth, I would suggest streamlining unnecessary details to minimize distraction for the reader.

Line 94-99, Page 4-5: What is the so-called promising initial empirical evidence? I wish the authors could be more specific in the content and not be vague.

Line 100-107, Page 5: Does the current content argue that ET is predictive of mental disorders? Or is there any other authoritative literature that apparently gives this conclusion?

Line 121-135, Page 6: Have the authors looked at the measurement properties of other language versions of the ETMCQ? This is important for the cross-cultural validation status of this scale.

Overall: The authors need further discretion in the background of the research, and providing relevant data would be a more intuitive approach. Also, the authors need to be more precise and cohesive in their presentation.

Methods

Participants and Procedures

1) The sample information needs to be further refined, e.g. from which region did the sample come? What were the inclusion and exclusion criteria for the sample? Is the sample representative of the general population?

2) Authors should also describe sample collection procedures, e.g. how was the sample size determined? How were the questionnaires distributed and collected? How was quality control performed, etc., whether in detail or briefly, I think this is necessary to describe a study’s methodology.

3) The sample size for Sample 2 is so small that the representativeness of the results is questionable.

Statistical Analysis

Line 301-314, Page 13: Effect size on outcome should be moved to the results section.

Results

1) I would suggest the authors to adjust the tables to a three line table format, also I think the font in the NOTE should be separate from the main text.

2) Only important results need to be described in text, otherwise what is the purpose of the table? This problem of redundancy is particularly evident in the section on Associations with demographic features.

3) The results of some of the model fits in Table 2 are not very ideal, did the authors consider further factorial constraints.

4) The figures in the appendix are missing titles and notes.

Discussion

1) On page 15, lines 337-338, the authors state that “compared to a representative sample of the German general population, sample 1 showed a higher burden”. What is the reason for such a difference?

2) Based on the main points and length of the authors’ discussion, I would suggest that the authors further refine their language, incorporate more of the relevant literature, and explain and extend inferences from the study’s findings. Honestly the content now makes it a struggle for me to read.

Implications and future directions

1) I’m not sure how the authors considered this part, however it is apparently that this section looks more like a discussion of the results than the impact of the outcomes of this study.

2) I don’t see any future research directions planned by the authors in this section, and in conjunction with the first sentence of the first paragraph, perhaps the authors are forgetting to add this content.

Conclusion

The conclusion of the study should provide an overview of the background, content and methodology of the study, followed by a list of the main conclusions, and finally a full-text summary or an outlook for future research. The current content is not a conclusion, but rather like a brief introduction. Authors need to read the relevant literature and learn how to write a competent conclusion.

Reviewer #2: The paper titled "Epistemic Trust and Associations with Psychopathology: Validation of the German Version of the Epistemic Trust, Mistrust, and Credulity-Questionnaire (ETMCQ)" aims to validate the self-report measure ETMCQ in the German population and explores the concept of epistemic trust and its potential associations with psychopathology. While the study certainly has some strengths in addressing an important area of research and contributes to the field of studies on epistemic trust and its dimensions, it has some notable weaknesses that warrant careful consideration.

The paper may contribute to cross-cultural comparison, as the ETMCQ was validated in different populations. Moreover, the author’s effort to use a transdiagnostic approach when discussing and studying ET and psychopathology is appreciable. However, the study presents some issues and lack of transparency regarding the factorial validity of the scale. These weaknesses currently hinder the publication's consideration. Further refinement and psychometric investigation of the data is vital to improve the paper.

Here the editor and the authors may find specific comments with respect to some aspects of the paper that need to be tackled. Addressing these issues would enhance the overall presentation and impact of the research findings.

Therefore, my main reservations concern the research design and, consequently, the method and result sections. In more details:

• In the manuscript, the authors refer to “the first investigation of the validity of the German version [29], partly deviating results of the factor structure compared to Campbell et al. [11]” (see page 6, lines 130-134). The reference of Nolte [29] is reported as follows: Nolte T. [German version of the Epistemic Trust, Mistrust and Credulity – Questionnaire]. Forthcoming. It appears to be another study validating the same instrument in the same language; hence, I believe it would be helpful if the authors can specify if and how their work differs from the previous one. More importantly, since they follow its proposed 12-item version for the ETMCQ, the reader should have access to the data of first paper for a comprehensive understanding. If the data is not publicly accessible yet, the authors should consider waiting until this paper is published. Furthermore, another point on which further clarification is needed is why the authors chose to use the 18-item version of the ETMCQ for their analyses when in the original validation of the instrument the items 16, 17, and 18 were excluded.

• Page 12, lines 280-283. In the paragraph of “Statistical Analysis”, the authors state that “In the third and fourth model, three items with the lowest factor loadings were removed and a 12-item version of the ETMCQ without and with correlated residuals was tested [29]”. In addition to the problems created by the unavailability of a readable reference to Nolte’s paper, which the authors repeatedly cite, it would be advisable to include the results of the PCA (or EFA), so that the factor loadings of each of the items are available to the reader. This is particularly crucial given that the Confirmatory Factor Analysis (CFA) reported by the authors appear to include items (i.e., item 16, 17, and 18) that were previously excluded from the ETMCQ, both in the original validation by Campbell and colleagues and in other currently published validations (e.g., the Italian one by Liotti and colleagues). Including PCA or EFA results would provide valuable insights into the factor structure and the retained items’ suitability for the German version of the ETMCQ.

• The use of standardized residuals in the CFA for a fairly new instrument like the ETMCQ should be approached with caution; it would be advisable for the authors to provide more support for their decision, especially since they end up validating a different version of the instrument than the original one (Campbell et al. 2021), as they used some item excluded in the original version.

• The results regarding construct validity should be described more thoroughly, and the authors should make assumptions about their significance based on the existing literature (for example, the relationship between epistemic mistrust and other certainty as measured by the CAMSQ, or the positive relationship between epistemic trust and negative affectivity could be discussed more exhaustively).

• The inter-scale correlation between epistemic mistrust (EM) and credulity (EC) is notably high. This result seems to indicate that there is a significant degree of overlap, or shared variance, between the two scales. Again, while it is possible to hypothesize that EM and EC may coexist in the same individual, the two ETMCQ subscales are theoretically expected to represent fundamentally opposite attitudes toward interpersonally transmitted information. It is not clear how the authors justify such a strong correlation. Moreover, it’s reasonable to assume that the coexistence of mistrust and credulity (i.e., two maladaptive stances) would be more likely to be found in psychopathological subjects. However, the sample in this study is purportedly normative. This finding warrants further discussion and examination throughout the article to better elucidate its implications and potential theoretical or methodological considerations.

• Moreover, in the method section the authors specify that for correlations they used Spearman and that they used Wilcoxon rank test for group differences. I think they should briefly clarify why they chose to use these non-parametric methods.

• The authors conducted a t-test to examine the difference between clinical and non-clinical samples with respect to the ETMCQ domains on 30 participants per group, which is considered the minimum sample size to conduct this analysis. Even if the differences between groups are not significative, I think the authors should have computed a power analysis (for example, using GPower), to estimate the effect size and specify what was the probability to find the effect, considering that the authors hypothesized its existence. The power analysis results should be reported. Moreover, given that with an independent samples t-test conducted on a few participants the minimal effect expected to be found is a large effect, the authors should specify (even if only in a footnote) that this might be the reason why this result is not significant. The need for research on the difference between clinical and non-clinical samples regarding epistemic trust may be explored in the Implications and future directions section.

Other critical issues that should be addressed are reported below.

Specifically, in the “Introduction” section:

• Pag. 3, lines 45-46. “Epistemic trust (ET) refers to an individual's willingness to perceive signals sent by others as relevant, trustworthy, and generalizable to other contexts”. Here I would say “transmitted” instead of “sent” and I would substitute “signals” with “information”, as this terminology seems more appropriate to describe epistemic trust rightfully.

• Page 3, lines 60-61. “Therefore, epistemic vigilance acts as a protective mechanism to question possible misinformation”. Here I would explain in more detail what epistemic vigilance is, since it is a pivotal concept for the theory of ET.

• Page 4, lines 69-74. Here there is some ambiguity regarding whether the authors are asserting that “feelings of isolation,” often linked with epistemic mistrust, subsequently lead to epistemic hunger, ultimately resulting in epistemic credulity. While it is plausible that individuals with elevated psychopathological tendencies and a history of adverse experiences (e.g., individuals with BPD) may exhibit a coexistence of both mistrust and credulity, it would be more precise to acknowledge that, regardless of whether they co-occur within the same individual, both mistrust and credulity are recognized in the literature on epistemic trust as maladaptive epistemic stances stemming from traumatic experiences, particularly during childhood (as subsequently elaborated upon in the article). The current phrasing of the sentence might inadvertently convey the idea that mistrust serves as the precursor to credulity through the mechanism of epistemic hunger.

• Pag. 7, lines 149–168. To the point about hypotheses and aims, it would be helpful if the authors provided further clarification on their expectations concerning the relationship between sociodemographic characteristics and the three subscales of the ETMCQ.

In the “Discussion” section of the manuscript:

• Page 22, lines 462-467. “The high correlative relationship between these two dimensions (mistrust and credulity) can be substantiated by the assumption that in the absence of ET, rapid shifts between credulity and hypervigilance could exist, therefore the two factors are closely related to each other. However, this interpretation raises the question of whether the mistrust subscale measures hypervigilance rather than persistent epistemic mistrust as a result of developmental adaptation to an adverse social environment”. The concept and term “epistemic mistrust” coincides with a rigid and all-encompassing epistemic hypervigilance, hence, the reason why the authors contrast mistrust with epistemic hypervigilance rather than epistemic vigilance is puzzling to me. Perhaps this section could benefit from further elaboration to clarify the rationale behind this comparison.

Overall, I would like to highlight the need for a proofreading or a thorough revision of the English language, as to improve clarity and readability.

6. PLOS authors have the option to publish the peer review history of their article (what does this mean?). If published, this will include your full peer review and any attached files.

Reviewer #1: No

Reviewer #2: No

---

## [Author Response · Author response to Decision Letter 0]

16 Jan 2024

All required answers to the reviewers and editors comments are included in the rebuttal letter that responds to each point raised by the academic editor and reviewer(s). This letter is uploaded as a separate file labeled 'Response to Reviewers'.

---

## [Decision Letter · Decision Letter 1]

4 Apr 2024

PONE-D-23-22407R1Epistemic trust and associations with psychopathology: Validation of the German version of the Epistemic Trust, Mistrust and Credulity-Questionnaire (ETMCQ)PLOS ONE

Dear Dr. Weiland,

Thank you once more for resubmitting your Manuscript. I think you did a great job in providing a very interesting research report, both from academic and applied perspectives.

Having taken on the role as Academic Editor of your revised version, I have recruited one additional reviewer since Reviewer 2 from the previous round did not respond anymore. Reviewer 1 did provide another review which overall maintained some of their earlier criticism. The new reviewer (Reviewer 3) was more specific in their commentary. Please find enclosed/attached the two reviews. I recommend to consider all the points carefully.  

Personally, I think your Manuscript is well written, and follows established scientific standards and practices. I support publishing this piece of work. Below I will specify what you would need to change before I can make a final judgement:

Can you please review the statistical comment made by Reviewer 3 – an expert on SEM –  about the skewed distributions of variables in the SEM? You may want to double-check with a statistician. Please modify as needed and/or provide evidence in the text or a supplement why skewness in the observed range is no problem for MLR. (Otherwise, page 13 is fine.)
All reviewers comment on the surprisingly high correlation of credulity and mistrust. It is particularly surprising because these two concepts have been introduced as “opposites”! In view of your findings, then, how can you speak of a “validation”? Especially since the Campbell data show a different structure. I like your interpretation about the “rapid shifts”,  but you need to explain why this should be the case only in your data. Could the elimination of the three low-loading items have led to that difference? What do the residual correlations say? (Again, consider a supplement). Can you check whether 2 or 3 factors are more suitable in your case? In fact, you speculate about this in the Discussion, so why not test the idea? Or test for invariance between your version and the one of Campbell? In whichever way you resolve the issue, please make sure this is being pointed out clearly in the Abstract. We want to ensure that readers of the Abstract understand that the underlying model of your scale is different from the original version.
In the Introduction, although you describe credulity and mistrust as “opposites”, you sometimes speak of “*positive* associations between mistrust, credulity, and [other variables]”. The phrase is difficult to reconcile with your claim that credulity and mistrust are antagonistic. Can you clarify? If these are opposites, yet both correlate with variable X, I would expect the text to say: “positive associations between variable X and mistrust on the one hand and credulity on the other”, or: “associations that are positive between variable X and mistrust, but negative between variable X and credulity”. Likewise, in line 91, speaking of opposites, I would expect you to say: “both mistrust and credulity are considered to be adaptive responses…” Same holds for Hypothesis 3c and d on page 7, and summary of findings in the discussion, e.g. line 501. In fact, your hypotheses truly give away that you presumed from the outset that these two cannot be opposites! BTW any time you’d like to double-check your English, I recommend using deepl, in case you haven’t tried it yet. I think your language is very good though.
As Reviewer 1 points out, the work of Csibra and Gergely can be left out on page 1 or described only very briefly or maybe moved further down in the Introduction to substantiate the choice of childhood trauma measures. Generally, could you please have a closer look at the structure of the text on page 1 (BTW an unusually long paragraph!). First, you define ET. Then you jump to  “natural pedagogy”, but without ever stating explicitly what this has to do with ET. [I think I know what you mean to say, but you don’t say it, and even if you did, it would disrupt the flow as you are leaving the level of conceptual definition and start speaking about ontology and adaptive functionality]. Then, starting with “However”, you suddenly go back to the level of conceptual definition. You say something along the lines of trust is not always beneficial, without saying why: Probably because it can open the door to being manipulated and exploited. If you’d say something like this explicitly, the following “therefore” would make much more sense! I hope you see what I mean… I am pointing this out so diligently only because the opening statements of a paper are so important.
Reviewers 1 and 3 were skeptical about study 2 for the low sample sizes. To address this point, you could perhaps state from the outset – perhaps on the basis of a power analysis – that your study design would allow you to detect only large effects. However, unlike Reviewer 3, I would not throw the study out, in the name of Open Science. After all, it is data; and you have made the attempt. Also, for that exact reason, I’d advise against calling it a “random result”. That phrasing seems to suggest that there was a positive finding that is based on chance. In reality, however, there no positive result and almost only chance. That’s why your only conclusion can be that you cannot interpret the pattern. Bayes analyses might help to see based on your data, how much more likely a null effect is compared to your hypothesized positive difference, but performing this analysis is not mandatory. However, please clearly state in the Abstract that you were “unsuccessful in finding” or “failed to find” (or something along those lines) the expected difference between the two groups. On a minor note, could you add one sentence in the Methods section about recruitment of the non-clinical sample?Could you add sample items to the description of the subscales of CAMSQ and LPFS?Line 253: Should be adverbial: “Exploratorily”Line 174: Should be “diagnosis”Table 2: Could you segregate model number from numbers of items further? Perhaps with a line break? à “Model 1: [line break] 15 items”

If you can address these points, I would be delighted to see the paper published in PLOS One soon.

Thank you once more for your thorough preparation of the revised Ms. I am hoping the above points provide you with a specific guideline for a second revision.

Kind regards,

Sabine Windmann

Academic Editor

PLOS ONE

Journal Requirements:

Reviewers' comments:

Reviewer's Responses to Questions

**Comments to the Author**

1. If the authors have adequately addressed your comments raised in a previous round of review and you feel that this manuscript is now acceptable for publication, you may indicate that here to bypass the “Comments to the Author” section, enter your conflict of interest statement in the “Confidential to Editor” section, and submit your "Accept" recommendation.

Reviewer #1: (No Response)

2. Is the manuscript technically sound, and do the data support the conclusions?

Reviewer #1: Partly

3. Has the statistical analysis been performed appropriately and rigorously? 

Reviewer #1: No

4. Have the authors made all data underlying the findings in their manuscript fully available?

Reviewer #1: Yes

5. Is the manuscript presented in an intelligible fashion and written in standard English?

Reviewer #1: Yes

6. Review Comments to the Author

**Reviewer #1:** ID: PONE-D-23-22407R1

Title: Epistemic trust and associations with psychopathology: Validation of the German version of the Epistemic Trust, Mistrust and Credulity-Questionnaire (ETMCQ)

Thank you for providing a chance to review this manuscript.

I have read the revised full text, and I have to say that this article is a very long story even though I understand the authors’ intention and core information, I still suggest, as I did in the first round, that the authors should streamline the language to lead the reader directly to the research topic and question and to cut out redundant information. The details of my question follow:

1. While the introduction effectively introduces the concept of ET and its relevance, some parts could be condensed for clarity and conciseness. For example, the discussion of Csibra and Gergely’s work on natural pedagogy could be summarized more succinctly.

2. If “epistemic mistrust” and “credulity” are distinct constructs or if they are part of a broader ET construct? Please clarify this. Please provide more detailed information about empirical studies supporting the discussed concepts, especially regarding the associations between ET, mistrust, credulity, and psychopathology. This would strengthen the credibility of the arguments presented.

3. Some citations are presented as footnotes ([1], [2]), while others are within the text ([5, 6]). Please choose one format and apply it consistently.

4. I remain of the opinion that the size and composition of Sample 2 may have affected the reliability and generalization of the results and were not sufficient to support what this study was trying to explore.

5. While you mention data cleaning, consider providing a brief overview of the specific steps taken to clean and prepare the data for analysis, as this can impact the validity and reliability of the results. Please consider including any robustness checks or sensitivity analyses conducted to ensure the robustness of the study findings.

6. I’m not very satisfied with the existing table presentation; the words are squished together, which detracts from the basic clarity.

7. In the discussion section, the authors mention the consistency and discrepancies between the findings and prior theories, but the reasons for these discrepancies could be explored in more depth. For example, the authors could have discussed the focus of different theoretical frameworks for explaining trust, mistrust, and gullibility behaviors, and whether these focuses explain the findings differently.

Overall, I consider this study to be open to scrutiny in terms of quality and needs significant revision before it can be published. I am sorry to reject the manuscript and look forward to future opportunities to see more of your excellent work.

Thank you and my best,

Your reviewer

7. PLOS authors have the option to publish the peer review history of their article (what does this mean?). If published, this will include your full peer review and any attached files.

Reviewer #1: No

---

## [Author Response · Author response to Decision Letter 1]

26 Aug 2024

All reviewers and editor comments are addressed in the "Response to reviewers"-Letter attached in this submission.

---

## [Editor Report · Decision Letter 2]

13 Sep 2024

PONE-D-23-22407R2Epistemic trust and associations with psychopathology: Validation of the German version of the Epistemic Trust, Mistrust and Credulity-Questionnaire (ETMCQ)PLOS ONE

Dear Dr. Weiland,

Thank you for submitting your revised manuscript to PLOS ONE. The Ms is acceptable in my opinion. You have addressed the comments raised.I am delaying the final decision only to provide you with the final opportunity to check on formal issues like language, text structure, and data visualization.

We look forward to receiving your revised manuscript.

Kind regards,

Sabine Windmann

Academic Editor

PLOS ONE
---

## [Author Response · Author response to Decision Letter 2]

15 Oct 2024

We addressed the final language and text structure checks in the cover letter, as there were no specific reviewer or editor comments in the decision letter.

---

## [Editor Report · Decision Letter 3]

17 Oct 2024

Epistemic trust and associations with psychopathology: Validation of the German version of the Epistemic Trust, Mistrust and Credulity-Questionnaire (ETMCQ)

PONE-D-23-22407R3

Dear Dr. Volkert,

We’re pleased to inform you that your manuscript has been judged scientifically suitable for publication and will be formally accepted for publication once it meets all outstanding technical requirements.

Kind regards,

Sabine Windmann

Academic Editor

PLOS ONE
---

## [Editor Report · Acceptance letter]

5 Nov 2024

PONE-D-23-22407R3 

PLOS ONE

Dear Dr. Volkert, 

I'm pleased to inform you that your manuscript has been deemed suitable for publication in PLOS ONE. Congratulations! Your manuscript is now being handed over to our production team.

Kind regards, 

on behalf of

Prof Sabine Windmann 

Academic Editor

PLOS ONE